# Short-Term ONX-0914 Administration: Performance and Muscle Phenotype in *Mdx* Mice

**DOI:** 10.3390/ijerph17145211

**Published:** 2020-07-19

**Authors:** Dongmin Kwak, Guoxian Wei, LaDora V. Thompson, Jong-Hee Kim

**Affiliations:** 1Department of Physical Therapy and Athletic Training, Boston University, Boston, MA 02215, USA; dkwak@bu.edu (D.K.); weigx@bu.edu (G.W.); 2Department of Physical Education, Hanyang University, Seoul 04763, Korea

**Keywords:** Duchenne muscular dystrophy, immunoproteasome, LMP7, ONX-0914, motor performance, inflammation, 7-week old *Mdx* mice

## Abstract

Duchenne muscular dystrophy (DMD) is a severe muscle-wasting disease. Although the lack of dystrophin protein is the primary defect responsible for the development of DMD, secondary disease complications such as persistent inflammation contribute greatly to the pathogenesis and the time-dependent progression of muscle destruction. The immunoproteasome is a potential therapeutic target for conditions or diseases mechanistically linked to inflammation. In this study, we explored the possible effects of ONX-0914 administration, an inhibitor specific for the immunoproteasome subunit LMP7 (ß5i), on motor performance, muscular pathology and protein degradation in 7-week old *MDX* mice, an age when the dystrophic muscles show extensive degeneration and regeneration. ONX-0914 (10 mg/kg) was injected subcutaneously on Day 2, 4, and 6. The mice were evaluated for physical performance (walking speed and strength) on Day 1 and 8. We show that this short-term treatment of ONX-0914 in *MDX* mice did not alter strength nor walking speed. The physical performance findings were consistent with no change in muscle inflammatory infiltration, percentage of central nuclei and proteasome content. Taken together, muscle structure and function in the young adult *MDX* mouse model are not altered with ONX-0914 treatment, indicating the administration of ONX-0914 during this critical time period does not exhibit any detrimental effects and may be an effective treatment of secondary complications of muscular dystrophy after further investigations.

## 1. Introduction

Duchenne muscular dystrophy (DMD) is a severe muscle-wasting disease that affects 1 in 3500 male births worldwide, which is caused by mutations in the X-linked gene encoding the cytoplasmic protein dystrophin [1]. Dystrophin is expressed in striated skeletal muscle and interacts with the dystrophin-glycoprotein complex at the cell membrane as well as specific signaling processes [2,3]. The lack of functional dystrophin results in progressive loss of sarcolemmal integrity, enhanced inflammation, increased fibrosis, chronic degeneration and increased proteolysis [1,4,5]. Currently, there is no cure for DMD, although various potential therapies are being tested (e.g., exon skipping, gene replacement, stem cell therapy, utrophin upregulation, read through therapy and endonuclease-based gene repair) [6].

Although the lack of dystrophin is the primary defect responsible for the development of DMD, secondary disease complications such as persistent inflammation and impaired regeneration contribute greatly to the pathogenesis and the time-dependent progression of muscle destruction [1,6,7]. To date, it is well-established that the dystrophic muscles are invaded by macrophages, neutrophils, and T-cells [8,9,10,11,12]. The infiltration of macrophages is a major source for the production of inflammatory cytokines/chemokines, which in turn recruit inflammatory cells and further promote inflammation (e.g., significant up-regulation of inflammatory genes and pathways) [13,14,15,16]. Glucocorticoids, which have anti-inflammatory properties, are used to treat the secondary muscle pathology in DMD; however, long term treatment is questionable because it induces muscle wasting and is accompanied by considerable side effects, such as hypertension, hyperglycemia, and osteoporosis [17]. Therefore, alternative therapies for regulating the immune response and slowing down disease progression in DMD patients are needed. In addition to glucocorticoids, other pharmacological strategies are being considered to mitigate the secondary and downstream pathologies [16,18,19]. Although each strategy has its own benefits, the administration of the intervention requires careful timing due to the disease progression [18,20].

Immunoproteasome is an alternative form of the constitutive proteasome [21]. It plays a key role in regulating the production of proinflammatory cytokines and maintaining protein homeostasis under cytokine-induced stress [21]. In skeletal muscle, the content of the immunoproteasome increases under conditions associated with inflammation, such as aging [22,23], denervation [24,25], stressful exercise [26] and muscular dystrophy [5,27]. Therefore, the immunoproteasome is a potential therapeutic target for conditions or diseases mechanistically linked to inflammation.

ONX-0914 is an inhibitor specific for the immunoproteasome subunit LMP7 (ß5i), which controls proinflammatory cytokine/chemokine expression, antigen presentation, infiltration, and tissue damage [28,29,30,31]. To date, ONX-0914 is effective in reducing the mRNA expression profiles of proinflammatory cytokine production, decreasing the levels of autoantibodies in modulating cytotoxic T cell responses and in reducing cellular infiltration in various mouse models of diseases such as myocarditis, colitis, diabetes, arthritis and *Candida albicans* infection, resulting in attenuation of disease progression [28,30,31,32,33]. Moreover, ONX-0914 treatment is a promising therapeutic approach for secondary pathological complications in dystrophic muscle. Specifically, when ONX-0914 treatment is initiated in 6-week old *MDX* mice and continues for five weeks, the results show a reduction of the number of CD45^+^ infiltrating cells in muscle tissue, a reduction in signs of degeneration (hypertrophic fibers, fiber splitting and fat replacement), and a diminished degree of fibrosis [27]. However, in the *MDX* mouse model, this time course of ONX-0914 administration (5 weeks) occurs over the entire enhanced regeneration and degeneration period [34]. Because of these promising results, the need to investigate protein expression and the time-dependent disease progression in the *MDX* mouse model further studies are desirable that focus on ONX-0914 [35]. Here, we aimed to explore the potential effects of short-term ONX-0914 administration, defined in this study as every two days for seven days, on motor performance and muscular pathology in 7-week old *MDX* mice, an age when the dystrophic muscles initially display extensive degeneration and regeneration.

## 2. Materials and Methods

### 2.1. Animals

Seven-week old male C57BL/10J mice (stock number: 000665), hereafter referred to as wild type (WT, *n* = 12) and dystrophin-deficient (*MDX*, *n* = 12) mice (C57BL/10ScSn-Dmd^mdx^/J; stock number 001801) were purchased from the Jackson Laboratory (Bar Harbor, ME). Mice were housed in groups (20–23 °C, 12-h photoperiod) and fed ad libitum. The study was approved by the Institutional Animal Care and Use Committee at Boston University.

### 2.2. Experimental Design

WT and *MDX* mice were divided into four experimental groups (WT-V: WT with vehicle treatment, WT-O: WT with ONX-0914 treatment, MDX-V: MDX-vehicle treatment, and MDX-O: MDX-ONX-0914 treatment). ONX-0914 (Selleckchem, Houston, TX, USA) was formulated in an aqueous solution of 10% (w/v) sulfobutylether-β-cyclodextrin and 10 mM sodium citrate (pH 6); whereas the vehicle treatment was a solution of 10 mM sodium citrate (pH 6) [29,32,36]. The duration of the experimental protocol was nine days with ONX-0914 (10 mg/kg) or vehicle administrated at 100 µL/20 g on Day 2, 4, and 6 (subcutaneous, s.c.). The duration and dose were based on previous reports that showed a sharp decline of LMP7 activity occurred at doses ranging from 1–10 mg/kg without apparent disturbance in other immuno-subunits’ activity in mice [28] and the effective administration interval ranged from one day to about one week [37]. Body weight, walking speed and strength were evaluated on Day 1 and 8. Muscle tissue was collected on Day 9 (Figure 1). In the current study, we used a selective immunoproteasome inhibitor because most of the non-selective proteasome inhibitors have issues with toxicities that limit their clinical application [38].

### 2.3. Walking Speed

On Day 1 and 8, the mice were placed on the Rota-Rod (PanLabLetica, Cornella, Spain) and walked at 4 rpm for 30 s [39,40]. Next, the rotation speed increased to 40 rpm over a 5-min period. When the mouse was unable to sustain the rotation speed, walking speed was recorded (in seconds). Each mouse performed three trials with a 10-min rest period between each trial. The best trial was selected for the outcome measure of walking speed. Because the body weight was different between WT and *MDX* mice, walking speed was also normalized to body weight.

### 2.4. Strength

The mice were placed on the grip meter grid (Coulbourn Instruments, Whitehall, PA, USA) and the tail was pulled back (torso kept in horizontal position) [39,40]. When the mouse was unable to maintain its grip, the trial ended, and the maximal grip strength was recorded (in grams). Each mouse performed two trials with a 10-min rest period between each trial. The best trial was selected for the outcome measure of grip strength. The grip strength was also normalized to body weight.

### 2.5. Muscle Tissue Harvest

On Day 9, mice were anesthetized with a mixture of ketamine and xylazine (100 mg/kg ketamine, 10 mg/kg xylazine). For histological experiments, the extensor digitorum longus (EDL) muscles were dissected, weighed, embedded in a tissue freezing medium, immediately frozen in nitrogen-cooled isopentane and stored at −80 °C. For proteasome subunit experiments (Immunoblotting), the gastrocnemius (GAS) muscles were dissected, weighed, immediately frozen in liquid nitrogen and stored at −80 °C. The tibialis anterior (TA) muscles were also dissected and weighed.

### 2.6. Histology

In order to determine the cross-sectional area (CSA) of single muscle fibers and the percentage of central nuclei (CN), histological cross-sections were prepared [5]. The frozen EDL muscles were sectioned at 10 µm (Leica CM3050S), processed with hematoxylin and eosin (H&E, ScyTek Laboratories, Lagan, UT, USA) and imaged at 10× (Nikon, Tokyo, Japan) with NIS-Elements software (Nikon, Tokyo, Japan). The CSA and CN of 100–150 fibers/muscle were quantified using Image J (National Institutes of Health, Bethesda, MD, USA).

In order to determine the immune infiltrates (inflammatory status) within each EDL muscle, CD45^+^ cells, a leukocyte marker, were quantified. Briefly, muscle cross-sections were washed with phosphate-buffered saline (PBS) for 5 min and fixed with 10% formaldehyde solution for 10 min. After fixation, the sections were washed, blocked with 5% bovine serum albumin (BSA), dissolved in PBS for 30 min and rewashed with PBS. The cross-sections were then incubated with a primary antibody (CD45, 1:30. R&D Systems, Minneapolis, MN, USA), diluted in 5% BSA dissolved in PBS for 1 h and washed as previously described [26]. The cross-sections were incubated with a secondary antibody (Alexa Fluor 488-conjugated antibody, 1:500, Life Technologies, Gaithersburg, MD, USA), diluted in 5% BSA dissolved in PBS for 30 min and rewashed. After the wash, the cross-sections were washed, dehydrated, mounted with a mounting medium containing 4′,6′-diamidino-2-phenylindole (DAPI, Abcam, Cambridge, MA, USA) and visualized at 20× (Nikon, Tokyo, Japan). Positively stained CD45 cells from 3 random fields per slide were quantified.

### 2.7. Proteasome Preparation

An enriched proteasome preparation was modified from those previously described [5,23,25]. Briefly, GAS (~150 mg) muscles were homogenized in buffer A (0.1 M KCl, 20 mM MOPS, pH 7.0) with Precellys Evolution homogenizer (Bertin Technologies, Montigny-le-Bretonneux, France). The homogenate was centrifuged at 4000× *g* for 20 min at 4 °C, and the supernatant was collected and saved. The remaining pellet was re-homogenized and re-centrifuged in the same way. The supernatant was then collected and combined with the initially saved supernatant. The pooled supernatant fractions were centrifuged at 12,000× *g* for 20 min at 4 °C. The supernatant was collected and centrifuged again at 100,000× *g* for 16 h at 4 °C. The resulting pellet from final centrifugation containing proteasome was suspended in buffer B (50 mM Tris-HCl, 5 mM MgCl_2_, 0.1% CHAPS, and 0.4% sucrose, pH 7.5) and stored at −80 °C. Protein concentration was determined with a bicinchoninic acid (BCA) assay (Thermo Scientific, Rockford, IL, USA) using bovine serum albumin (BSA) as a standard.

### 2.8. Immunoblotting

An aliquot of the extraction was mixed with Laemmli Sample Buffer (Bio-Rad, Hercules, CA, USA) containing 10 mM DTT (BIO-RAD) and heated at 95 °C for 5 min. Equal amounts of protein (20 µg for α7 and LMP7, 40 µg for LMP2) were then loaded onto 4%–15% Mini-PROTEAN^®^ TGX Stain-Free™ Protein Gels and separated by electrophoresis at 100 V for ~1 h using a mini-vertical gel electrophoresis unit (BIO-RAD). The proteins were transferred to nitrocellulose membrane using Trans-blot SD semidry transfer cell (BIO-RAD) at 14 V/blot for 45 min. For the determination of total proteins, the blots were stained with REVERT™ Total Protein Stain and Wash Solution Kit (LI-COR Biosciences, Lincoln, NE, USA) and imaged at 700 nm channel (red) with Odyssey imaging system. The blots were subsequently blocked in Odyssey Blocking Buffer-TBS (LI-COR Biosciences) at room temperature for 1 h and then incubated with primary antibodies in TBST buffer containing 0.2% Tween 20 overnight at 4 °C. After washing with TBST containing 0.1% Tween 20 (3 × 5min), the blots were probed with secondary antibody IRDye800CW goat anti-mouse (LI-COR Biosciences) and imaged at 800 nm channel (green) with Odyssey imaging system (LI-COR Biosciences). Before immunoblotting, the sample loading amount and conditions for each antibody were optimized to ensure that the reaction was within the linear ranges for signal intensity. Table 1 shows the protein load and antibody information.

Densitometry analysis was performed using the Odyssey infrared imaging system application software (LI-COR Biosciences, ver 3.0). Quantification was performed by total protein normalization methods with two equations: (1) Lane Normalization Factor (LNF) = Signal/Signal for the lane with highest Signal; (2) Normalized Signal = Target Band Signal/Lane Normalization Factor (Appendix A). A GAS muscle from a *MDX* mouse was used as a blot control to compare samples across different blots. The final protein content of each sample was expressed as a ratio to the blot control (AU). Six independent Western blot experiments were performed.

### 2.9. Statistical Analysis

All values were expressed as mean ± standard error. First, to determine genotype differences at seven weeks of age between WT and *MDX,* independent *t*-tests were used. An independent *t*-test was then used to determine differences between short-term Vehicle- and ONX 0914-treatment within each mouse strain on inflammatory status, strength, walking speed, muscle weight, the protein content of α7, LMP2 and LMP7, CSA, and CN. Statistical significance was defined as *p* < 0.05. All statistical analysis was performed using SigmaPlot version 14.0 (Systat Software, San Jose, CA, USA).

## 3. Results

### 3.1. Phenotype Characteristics of WT and MDX Mice at Seven Weeks of Age

To evaluate the phenotypic differences between 7-week old WT and *MDX* mice, we compared body weight, strength, walking speed, muscle immune infiltrates and the presence of muscle regeneration within the EDL. At seven weeks of age, the *MDX* mice weighed 7% more than WT mice (*p* = 0.008) (Figure 2A). In contrast, the strength and walking speed of the *MDX* mice were 22% and 56% lower than WT mice, respectively (*p* < 0.006) (Figure 2B,C). Because the *MDX* mice weighed more than WT mice at this age, strength and walking speed were normalized to body weight. The normalized strength and walking speed of *MDX* mice were also significantly lower than WT mice (*p* < 0.003) (Figure 2D,E). Because previous reports indicate that 8-week old *MDX* mice have extensive muscle inflammation and regeneration [41], the percentage of CD45^+^ cells (an immune infiltrate/inflammation marker) and CN (a regeneration marker) was determined in both *MDX* and WT mice. The percentage of CN in the EDL muscle was significantly greater in *MDX* mice compared to WT mice (*p* < 0.001) (Figure 2F,G) at the age of 7 weeks. The percentage of CD45^+^ cells in the EDL muscle was 3-fold greater in *MDX* mice (*p* < 0.001) (Figure 2F,H).

### 3.2. ONX-0914 Treatment In Vivo: WT Mice

First, in order to identify the possibility that the ONX-0914 injection in vivo and/or the injection itself has an effect in healthy WT mice at seven weeks of age, we evaluated body weight, strength, walking speed, muscle weight, single muscle fiber CSA and the percentage of CD45^+^ cells and CN in the EDL muscle (Table 2). We also measured the proteasome content in the GAS muscle following a 9-day treatment period (Appendix A). As we expected, no statistical differences in body weight (*p* = 0.062), physical performance (*p* > 0.240), muscle characteristics (*p* > 0.548), muscle proteasome content (*p* > 0.446) and muscle immune infiltrates (*p* = 0.569) were observed between ONX-0914 treated WT mice and WT mice treated with vehicle (Table 2 and Appendix A proteasome and subunits). These data suggest that the ONX-0914 treatment in vivo does not have a negative impact in WT mice.

### 3.3. ONX-0914 Treatment In Vivo: MDX Mice

In order to elucidate the role of ONX-0914 treatment in vivo for a period of nine days in *MDX* mice, starting at seven weeks of age and ending at eight weeks of age, we determined the inflammatory status, physical performance, muscle characteristics and proteasome content (Figure 3).

Inflammatory Status: We determined the inflammatory marker CD45^+^ cells because ONX-0914 treatment in vivo for five weeks is associated with reduced muscle immune infiltrates in *MDX* mice [27]. We found no differences between Vehicle- and ONX-0914-treated *MDX* mice in vivo (*p* = 0.774) (Figure 3A). These data indicate that ONX-0914 treatment for a period of nine days does not influence this muscle immune infiltrate.

Physical Performance and Muscle Mass: To determine whether the ONX-0914 treatment in vivo for a period of nine days impacts physical performance, we determined strength and walking speed in *MDX* mice. Strength and walking speed were not different between the Vehicle- and ONX-0914-treated *MDX* mice (*p* > 0.441) (Figure 3B,C). To further investigate these findings of no change in physical performance, we determined the muscle mass of both the TA and EDL. Similar to the findings in physical performance, the TA and EDL muscle mass did not differ between Vehicle- and ONX-0914 treated in vivo *MDX* mice (*p* > 0.193) (Figure 3D,E).

Proteasome Content: Because long-term ONX-0914 treatment (5 weeks) has been shown to influence proteasome properties in dystrophic muscle [27], we sought to evaluate the content of the standard proteasome and immunoproteasome subunits (Figure 3F,H). First, the α-subunits (α1–7) are constitutively expressed in both the standard proteasome and in the immunoproteasome; therefore, it is a reliable measure for total proteasome content [25,26]. We found that the content of α7 in the GAS muscles did not differ between Vehicle and ONX-0914 treated *MDX* mice (*p* = 0.700) (Figure 3F). Next, in the GAS muscle, we determined the inducible subunits (LMP2 and LMP7), and we found no statistical differences in the content of these subunits between the Vehicle and ONX-0914 treatment groups in MDX mice (*p* > 0.180) (Figure 3G,H).

CSA and regeneration (CN): We investigated the impact of ONX-0914 administration on single fiber CSA and the percentage of CN in the EDL muscle of MDX mice (Table 3). Consistent with the proteasome, immunoproteasome and subunit content, there was no statistical difference in the CSA nor in the percentage of CN between Vehicle- and ONX-0914-treated mice (*p* > 0.063) (Table 3).

## 4. Discussion

There are reports indicating that regulating the immune response has the potential to attenuate the disease progression in DMD, providing a multi-prong approach to treat the secondary pathological complications associated with this devastating muscle-wasting disease [6,7]. Therefore, in the current study, we aimed to explore the potential effects of short-term treatment of an immunoproteasome inhibitor, ONX-0914, on motor performance, muscle pathology and muscle degradation in 7-week old *MDX* mice, an age when there is enhanced cycles of degeneration and regeneration. We show that this short-term treatment of 10 mg/kg in *MDX* mice every other day for nine days did not alter strength nor walking speed. These findings were consistent with no change in muscle inflammatory infiltration, central nuclei and proteasome content. Importantly, short-term ONX-0914 treatment did not reduce motor performance nor worsen muscle pathology. Taken together, muscle structure and function in the young adult *MDX* mouse model are not altered with ONX-0914 treatment, indicating the administration of ONX-0914 does not exhibit any detrimental effects during this critical time period and may be an effective treatment of secondary complications of muscular dystrophy after further investigations.

ONX-0914 is an inhibitor specific for the immunoproteasome subunit LMP7 (ß5i) [28], which predominantly blocks LMP7-mediated protein hydrolysis [28,42]. To date, the promising therapeutic potential of targeting the immunoproteasome with ONX-0914 is reported for the treatment of various autoimmune diseases, including rheumatoid arthritis [28], systemic lupus erythematosus [43], encephalomyelitis [37] and for severe acute viral infections, such as myocarditis and rhinovirus infection [31,44]. Recently, the benefits of this treatment are seen in uterine contractility associated with preterm labor [45]. The premise for using ONX-0914 is based on the highly regulated immunoproteasome function of controlling proinflammatory cytokine/chemokine expression and antigen presentation, less infiltration and reduced tissue damage [28,29,30,31,37,46].

ONX-0914 treatment is a promising therapeutic approach for dystrophic pathology in the dystrophin-deficient *MDX* mouse. The *MDX* mouse is the classical pre-clinical animal model of DMD for investigating the disease progression because these mice exhibit a muscle pathology similar to humans between 3 and 8 weeks of age [1,7]. Indeed, a critical period in the disease progression is identified between the ages of 3 and 8 weeks, such that an acute onset of muscle necrosis is observed at three weeks of age followed by a 7-week acute phase of intensive myofiber degeneration and regeneration [1,7]. Infiltration of damaging inflammatory cells is also prevalent during this acute phase of degeneration and regeneration [8,9,10,11,12]. This age-dependent disease severity in *MDX* mice provides a good model to investigate interventions designed to prevent secondary pathological complications, such as inflammation. The promise of ONX-0914 as a treatment for the secondary pathological complications is seen when the treatment is initiated in 6-week old *MDX* mice and lasts for five weeks (two injections per week of ONX-0914 at 6 mg/kg) [27]. The long-term administration of ONX-0914, with *MDX* mice studied at 12 weeks of age, results in a reduction of the number of CD45+ infiltrating cells in muscle tissue, a reduction in signs of degeneration (hypertrophic fibers, fiber splitting and fat replacement), and a diminished degree of fibrosis. Importantly, the long-term treatment of 6 mg/kg did not result in tumor formation or delay in muscle differentiation; however, the treatment time course concurred across the full time period of degeneration and regeneration in the *MDX* mouse model [27]. Based on these encouraging findings for ONX-0914, the next step is to evaluate whether the administration of ONX-0914 is beneficial during the acute phase of intensive myofiber degeneration and regeneration, a period of increased inflammation, and to determine if the previously reported gene expression profiles result in protein alterations [8,9,10,11,12].

In order to evaluate whether the administration of ONX-0914 is beneficial during the acute phase of intensive myofiber degeneration and regeneration, the initial step in the current study required the identification of the motor function and morphological characteristics of muscle in *MDX* mice at seven weeks of age. The findings corroborate the evidence that in 7-week old *MDX* mice, there are clear motor deficits and the muscles are present with bouts of myofiber degeneration/regeneration [7,10,12,47,48]. The reported 59% centrally placed nuclei within the myofibers are in line with lifespan studies demonstrating an enhanced regenerative state at this age [49,50]. The muscles also present with the classical hallmarks of inflammatory infiltration at this age [11,12]. The dystrophic muscle phenotype of the *MDX* mouse at this early age is likely a major contributing factor in the observed loss of grip strength and walking speed.

Next, we explored the therapeutic benefits of short-term administration of ONX-0914 because we suggest the suppression of immunopathology achieved by immunoproteasome inhibition attenuates the disease progression. First and as expected, ONX-0914 injections do not result in any adverse reactions in healthy mice. Seven-week old ONX-0914 treated WT mice were in overall good condition as evidence by strength, walking speed, body weight, muscle weight and the cellular integrity of the muscle. In turn, there are no overall unwanted reactions to the dose nor to the short-term treatment timeline in *MDX* mice at seven weeks of age. The stable body weight and physical performance findings between the two treatment groups (ONX-0914 treated or Vehicle treated *MDX* mice) are supported by the muscle histopathology, the total proteasome contents, and the inducible subunits of the immunoproteasome, LMP2 and LMP7 findings. Therefore, it is possible to conclude that administration of ONX-0914 during this critical period of muscle remodeling does not cause further secondary muscle pathology (e.g., immune cell infiltration and enhanced degradation.)

Yet, we acknowledge enhancement of motor function and attenuation of muscle pathology did not occur at this age in *MDX* mice either. This finding is expected because it is likely the impact of short-term administration of ONX-0914 initiates a cascade of molecular and cellular events, such as Ikβ increase and deactivation of NF-kβ, which impact muscle structure and function in a time-dependent manner. In order to capture potential benefits in physical performance, gene and protein expression profiles and muscle morphology, future investigations focused on several time-points post-administration of ONX-0914 (e.g., > than 4–8 days post-administration of ONX-0914) in the *MDX* mouse model are needed. Likewise, the selection of these time-points requires careful consideration because of the regenerative capacity in the *MDX* mouse model [20]. Furthermore, in the current study we did not evaluate mRNA levels of these pathways and we assumed, based on previous short-term ONX-0914 treatment investigations, the treatment (dosage and timing) was effective at altering the pro-inflammatory cytokine gene expression profiles [28,30,31,36]. In this regard, further time-course studies will be required to fully address whether short-term administration of ONX-0914 during this acute phase of continual influx of inflammation, repeated cycles of degeneration, and impaired regeneration influences molecular and cellular pathways (e.g., gene expression signatures) and protein expression profiles, resulting in physical performance improvement [1,13,14,15]. Understanding the changes in molecular pathways secondary to short-term ONX-0914 treatment will assist in future experimental design.

Notably, in accordance with the current findings, we previously demonstrated an increase in the content of LMP7 and LMP2 and proteasome activities in dystrophic muscles of *MDX* and DKO (dystrophin- and utrophin-double-knockout) mice within the same age group [5]. Taken together, these data and previous studies suggest that the immunoproteasome is involved in maintaining cellular homeostasis [21,26]. They also lay the foundation to spur further exploration of the role of the immunoproteasome.

## 5. Conclusions

In closing, we demonstrate that administration of ONX-0914 in 7-week old *MDX* mice for a short duration period does not reduce motor performance nor worsen muscle pathology. Because this protocol for ONX-0914 treatment does not exhibit any detrimental effects, further investigations are needed to determine the impact of this protocol on the molecular and cellular pathways, which ultimately influence the extent of the secondary complications of muscular dystrophy.

## Figures and Tables

**Figure 1 ijerph-17-05211-f001:**
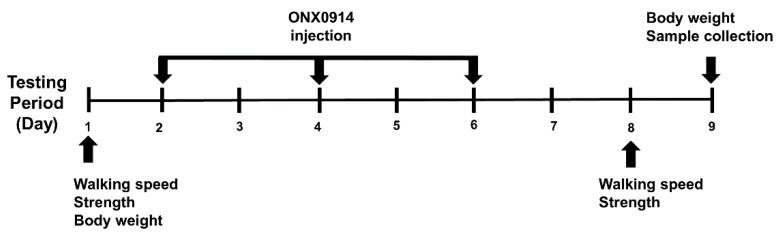
Experiment timeline. The duration of the testing period was nine days. The mice were evaluated for physical performance (walking speed, strength) on Day 1 and 8. Body weight was measured on Day 1 and 9. ONX-0914 (10 mg/kg) was injected subcutaneously on Day 2, 4 and 6. Muscle tissue was collected on Day 9.

**Figure 2 ijerph-17-05211-f002:**
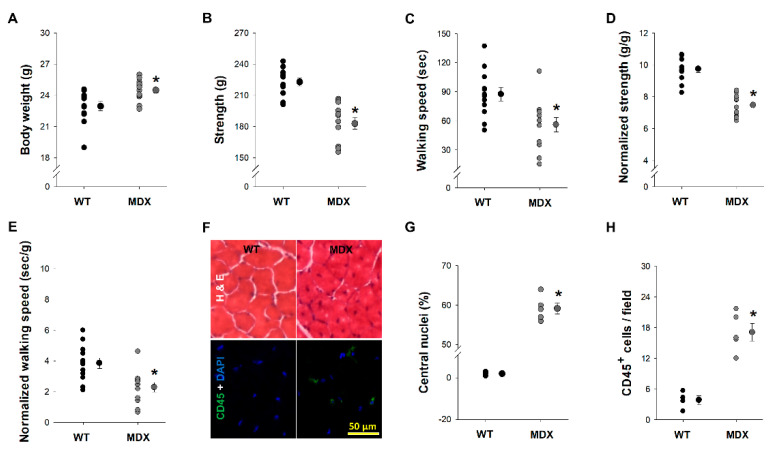
Characteristics of WT and *MDX* mice at seven weeks of age. Body weight (**A**), strength determined by the grip meter test (**B**), walking speed determined by the Rota-rod (**C**), normalized strength (**D**) and normalized walking speed (**E**) were evaluated in WT (black filled circles) and MDX (grey filled circles) mice. Data are presented as individual points (each mouse) with the mean and standard error of the means (S.E.M.). Representative images of hematoxylin & eosin, DAPI and CD45^+^ cells at 20× magnification are shown in (**F**). The mean percentage of central nuclei (**G**) and CD45^+^ cells per field (**H**) in EDL muscles of WT (black filled circle) and MDX (grey filled circle) mice were quantified. * indicates *p* < 0.05 comparing WT to MDX mice. Sample size: for body weight and physical performance, *n* = 12, for histology, *n* = 4–6 in each experimental group.

**Figure 3 ijerph-17-05211-f003:**
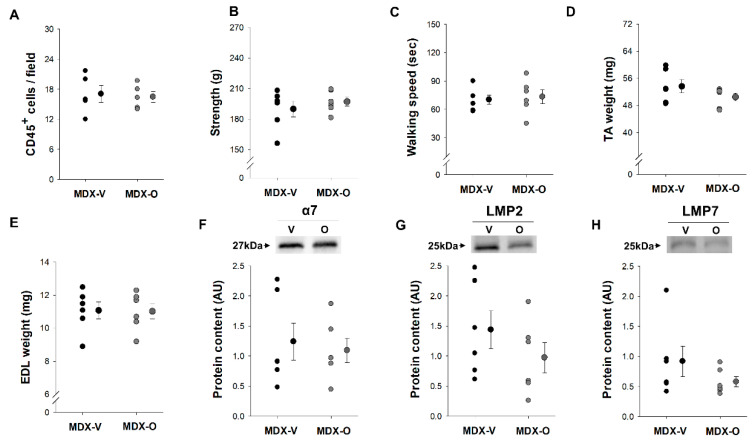
ONX-0914 treatment in *MDX* mice at seven weeks of age. The percentage of CD45^+^ cells in EDL muscles (**A**) of MDX mice treated with Vehicle (MDX-V, black filled circle) and treated with ONX-0914 (MDX-O, grey filled circle) were determined. Strength was determined by the grip meter test (**B**), walking speed was determined by the Rota-rod (**C**), and tibialis anterior muscle weight (TA, **D**) and EDL weight (EDL, **E**) in MDX-V and MDX-O mice were evaluated. The content of α7 (**F**, representative image), LMP2 (**G**, representative image), and LMP7 (**H**, representative image) was determined using Western blot analysis in the gastrocnemius muscles of MDX-V and MDX-O mice. Proteins were normalized to total protein (Appendix A) and expressed as an arbitrary unit (AU). Data are presented as individual points (each mouse) with the mean and standard error of the means (S.E.M.). * indicates *p* < 0.05 comparing Vehicle to ONX-0914 treatment groups. Sample size: *n* = 5–6 in each group.

**Table 1 ijerph-17-05211-t001:** Antibodies used for Immunoblotting.

Target Proteins	Protein Load (µg)	Primary Antibody		Secondary Antibody	
		Host	Dilution	Host	Dilution
α7	20	M	1:1000	M	1:15,000
LMP2	40	M	1:500	M	1:15,000
LMP7	20	M	1:500	M	1:15,000

All antibodies were isotype IgG. Host species mouse (M); All monoclonal primary antibodies were purchased from Enzo Life Sciences, Inc., Farmingdale, NY, USA; Secondary antibody: IRDye800CW goat anti-mouse (LI-COR Biosciences, Lincoln, NE, USA).

**Table 2 ijerph-17-05211-t002:** ONX-0914 treatment for a period of 9 days in WT mice.

Parameters	WT-Vehicle (7 Weeks)	WT-ONX (7 Weeks)
Body weight (g)	24 ± 0.6	22 ± 0.8
TA weight (mg)	42 ± 1.2	38 ± 0.7
EDL weight (mg)	8.8 ± 0.4	9.1 ± 1.5
Strength (g)	228 ± 5	221 ± 6
Walking speed (sec)	102 ± 11	126 ± 22
EDL muscle		
CSA (µm^2^)	1185 ± 104	1077 ± 32
Central nuclei (%)	2.0 ± 0.3	1.4 ± 0.2
CD45^+^ cells (number/field)	3.8 ± 0.8	3.3 ± 0.3
Ratio of TA weight to body weight (mg/g)	1.7 ± 0.04	1.7 ± 0.04
Ratio of EDL weight to body weight (mg/g)	0.4 ± 0.02	0.4 ± 0.06
Ratio of strength to body weight (g/g)	9.5 ± 0.4	10.0 ± 0.4
Ratio of walking speed to body weight (sec/g)	4.3 ± 0.6	5.7 ± 1.0

Data are presented as mean ± SEM. * *p* < 0.05 as compared with the corresponding WT with ONX-0914 treatment. For body and muscle weight, *n* = 6 per group; for CSA and central nuclei 100–150 individual fibers analyzed per animal and for CD45^+^ cells three random fields of EDL muscle at 20X, *n* = 4–5 per group.

**Table 3 ijerph-17-05211-t003:** ONX-0914 treatment for a period of 9 days in *MDX* mice.

Parameters	MDX-Vehicle (7 Weeks)	MDX-ONX (7 Weeks)
Body weight (g)	25 ± 0.3	24 ± 0.4
TA weight (mg)	54 ± 2.0	51 ± 1.2
EDL weight (mg)	11.1 ± 0.5	11.0 ± 0.5
Strength (g)	190 ± 8	197 ± 4
Walking speed (s)	70 ± 5	73 ± 7
EDL muscle		
CSA (µm^2^)	1237 ± 54	1111 ± 31
Central nuclei (%)	59 ± 1.4	58 ± 1.9
CD45^+^ cells (number/field)	17 ± 0.8	17 ± 1.1
Ratio of TA weight to body weight (mg/g)	2.2 ± 0.07	2.1 ± 0.04
Ratio of EDL weight to body weight (mg/g)	0.4 ± 0.02	0.5 ± 0.02
Ratio of strength to body weight (g/g)	7.7 ± 0.3	8.2 ± 0.2
Ratio of walking speed to body weight (s/g)	2.8 ± 0.2	3.0 ± 0.3

Data are presented as mean ± SEM. * *p* < 0.05 as compared with the corresponding MDX with ONX-0914 treatment. For body and muscle weight, *n* = 6 per group; for CSA and central nuclei 100–150 individual fibers analyzed per animal and for CD45^+^ cells three random fields of EDL muscle at 20×, *n* = 4–5 per group.

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
