# Peer review of "Short-Term ONX-0914 Administration: Performance and Muscle Phenotype in *Mdx* Mice"

_ijerph, 2020, doi:10.3390/ijerph17145211_

Round 1
Reviewer 1 Report
Title: Short-term immunopoteasome inhibition: Performance and muscle phenotype in Mdx mice
Summary:
This is an original study examining the effects of ONX-0914 administration, an inhibitor specific for the immunoproteasome subunit LMP7 (ß5i), on motor performance and muscular pathology and degradation in 7-week old MDX mice, an age when the dystrophic muscles show extensive degeneration and regeneration.
The authors reveal that inhibition of of immunoproteasome subunit MKP7 can protect skeletal muscular strength and walking performance in DMD.
Study design is appropriate and well designed to examine the study questions. Authors presented high quality of images and results. Additionally, conclusion is well represent their findings.
Major comments:
- Although this study is well designed and performed, authors should discuss about the effects of selective vs. non- selective immunoproteasome inhibitor on proteasome activity and level.
- Authors use only gastrocnemuius, is there any rationale that authors did not use soleous muscle?
- If authors can provide a comparison between soleus vs. gastrioc muscle, it will make study more comprehensive. If these data are not available, please discuss these muscle group dependent proteasome activity in the experimental consideration.
- I know authors used a local injection, however, did authors find any systemic changes in inflammation and/or oxidative stress levels after serial injection of ONX-0914?
- Authors reported skeletal muscle strength and walking performance, however, it is hard to understand this improvement is ONX-0914 mediated skeletal muscle strength improvement or associated with any other signaling mechanism such as neural effects. Is there any evidence that immunoproteasome inhibition can increase single-muscle fiber tension or tone? Please discuss about this in the discussion.
Author Response
May 17, 2020
Dear Reviewers and Editor,
We appreciate the careful attention and constructive suggestions by the reviewers for our manuscript, titled “Short-term Immunoproteasome Inhibition: Performance and Muscle Phenotype in Mdx Mice.”
To be clear, we are confused, and we are at a quandary of what steps to take next and the direction to go with the manuscript. The predicament: the comments by the two reviewers are at odds with each other. For example, reviewer 1 states “Study design is appropriate and well designed to examine the study questions. The authors presented a high quality of images and results. Additionally, the conclusion from findins is well represented.” In contrast to reviewer 1, reviewer 2 first states that “The present work by Kwak et al. is well presented and uses standard MDX mouse model outcome measures for the short-term assessment of subcutaneously delivered 10 mg/kg ONX-0914.” Subsequently, the reviewer 2 follows up with an opposing statement: “the fact that the shorter-term treatment schedule and alternative delivery mechanism (SubQ), used in this study, fails to achieve similar immunoproteasome inhibition suggests a study design or experimental implementation flaw.”
Because of the opposing messages between the two reviewers and the contrasting messages within reviewer 2, we have tried to fully explain below possible reasons for the confusion by reviewer 2. We have also made changes within the manuscript.
Indeed, there is a published study by Farini et al., 2016, by which ONX-0914 was administered for a long period of time, 5 weeks, beginning when the MDX mice were 6 weeks old and ending when the mice were 11 weeks old. Foremost, we use this published study as one of our primary rationale to investigate the benefits of “short-term treatment of ONX-0914”. We cite this published paper in the manuscript, and the major finding supports the use of ONX-0914 for a long period of time as a potential treatment for secondary pathologies associated with muscular dystrophy. In contrast to the positive results with long-term treatment of ONX-0914, short-term treatment of ONX-0914 did not result in positive results; however, the treatment was not detrimental. We clearly state and conclude that ONX-0914 treatment for a short-term duration in MDX mice at 6 weeks of age does not exhibit any detrimental effect on muscle pathology nor performance; however, the treatment does not have a significant beneficial effect in the MDX muscle disease progression. We believe we interpreted our findings according to our results of a well-designed study. Unfortunately, we believe that reviewer 2 views “the lack of a positive result” as a study design flaw. We respectfully disagree, well-designed studies do not always result in positive changes.
Below, we have done our best to answer each concern of the reviewers, based on our interpretation. We used red font in the manuscript to indicate changes to the text. Below, for each one of the respective reviewer’s concern (blue font) we provide specific responses (black font.)
If there are things that we did not address/answer to your satisfaction, please let us know. We appreciate your time and effort.
Sincerely,
Drs. Dongmin Kwak, Guoxian Wei, LaDora V. Thompson, and Jong-Hee Kim
Reviewer #1
Summary: This is an original study examining the effects of ONX-0914 administration, an inhibitor specific for the immunoproteasome subunit LMP7 (ß5i), on motor performance and muscular pathology and degradation in 7-week old MDX mice, an age when the dystrophic muscles show extensive degeneration and regeneration.The authors reveal that inhibition of immunoproteasome subunit MKP7 can protect skeletal muscle strength and walking performance in DMD. Study design is appropriate and well designed to examine the study questions. Authors presented high quality of images and results. Additionally, conclusion is well represent their findings.
Major comments:
- Although this study is well designed and performed, authors should discuss about the effects of selective vs. non- selective immunoproteasome inhibitor on proteasome activity and level.
Response: Important point. Indeed, previous research used non-selective proteasome inhibitors, which inhibited proteasome chymotrypsin-like activities through both the β5 and LMP7 subunits (Bakers et al, 2005, Chauhan et al, 2005, Demo et al, 2007). However, the non-selective proteasome inhibitors have problems with toxicities, which limit their use as therapeutics (Ettari et al. 2018).
In the current study, we used a selective immunoproteasome inhibitor of LMP7 (ONX-0914) because of its success in a previous study that investigated the long-term treatment in MDX mice (Farini et al., 2016).
To acknowledge the importance of the previous work with non-selective proteasome inhibitors we added the following information to the methods (line 90-92, page 2).
CHANGE IN MANUSCRIPT: “In the current study we used a selective immunoproteasome inhibitor because most of the non-selective proteasome inhibitors have issues with toxicities that limit their clinical application (16).”
- Authors use only gastrocnemuius, is there any rationale that authors did not use soleous muscle?
Response: In the current study, we used extensor digitorum longus (EDL), gastrocnemius (GAS), and tibialis anterior (TA) for experimentation/data analyses. In the manuscript, lines 112-117, page 3 describes the muscle harvest selection. These muscles are mainly dominated by type II fibers. We excluded the soleus muscle because it is a mixed fiber type, both type I and type II fibers.
- If authors can provide a comparison between soleus vs. gastrioc muscle, it will make study more comprehensive. If these data are not available, please discuss these muscle group dependent proteasome activity in the experimental consideration.
Response: We understand the reviewer’s perspective of wanting a comparison between gastrocnemius and soleus muscles. As we set up the study, we selected the type II muscles because the majority of studies using the MDX mouse model investigated type II muscles such as TA and quadriceps (Farini et al. 2016) and EDL (Chen et al. 2014), which are preferentially affected to muscle pathology and degeneration compared with type I muscles(Talbot et al. 2016, Glasser et al., 2018). Unfortunately, we did not collect soleus muscles. It is possible to add a comment about this in the manuscript; however, there is no physiological reason to hypothesize the soleus would respond significanlty than the gastrocnemius within the study design.
- I know authors used a local injection, however, did authors find any systemic changes in inflammation and/or oxidative stress levels after serial injection of ONX-0914?
Response: This is an interesting question. The concept fits nicely with the published papers when ONX-0914 is given as an intervention following the induction of a rapid, acute inflammatory response (e.g, Dimausuay et al, 2018). First, we did not obtain any plasma and/or serum samples from these young wild-type and MDX mice. Second, we did not evaluate systemic changes in inflammation nor oxidative stress levels after short-term ONX-0914 treatment in muscle. However, if the ONX-0914 short-term treatment had shown a beneficial effect in the MDX muscle disease progression and motor function, it would have been logical to evaluate skeletal muscle protein damage, degeneration, regeneration, etc. As a consequence of the findings, we did not pursue this.
- Authors reported skeletal muscle strength and walking performance, however, it is hard to understand this improvement is ONX-0914 mediated skeletal muscle strength improvement or associated with any other signaling mechanism such as neural effects. Is there any evidence that immunoproteasome inhibition can increase single-muscle fiber tension or tone? Please discuss about this in the discussion.
Response: Perhaps there is confusion with our results. Our data indicated that muscle strength and walking speed did not improve in MDX mice after short-term ONX0914 treatment. See results (lines 248-254, page 7). With this finding, it did not seem logical to perform single fiber contractility experiments. We are hesitant to speculate on whole muscle or single fiber contractility when it is likely contractility did not change based on performance and muscle characteristics. Note, we do acknowledge within the manuscript, the motor performance and the muscle characteristics following short-term ONX-0914 treatment are consistent.
Reviewer 2 Report
The present work by Kwak et al. is well presented and uses standard mdx mouse model outcome measures for the short-term assessment of subcutaneously delivered 10 mg/kg ONX-0914. The 7-week time point is within the dystrophic developmental period, where multiple rounds of degeneration/regeneration are taking place in this mouse model, prior to stabilization at 10 weeks of age. Unfortunately, there are several reasons this study will have a significantly diminished impact on the muscular dystrophy field. First, as referenced by the authors (Farini et al. Ref#24), a longer-term study using ONX-0914 has been previously conducted in the mdx mouse model, finding several beneficial effects and clear activity on the immune presence in skeletal muscle. Secondly, the fact that the shorter-term treatment schedule and alternative delivery mechanism (SubQ), used in this study, fails to achieve similar immunoproteosome inhibition suggests a study design or experimental implementation flaw. Therefore, it is difficult to draw conclusions from the lack of change between Vehicle and ONX-0914 in the study, due to a lack of on-target pharmacological activity. Due to this lack of activity, some of the conclusions discussed are unwarranted as they are based upon the presumption of ONX-0914 activity. For example, why would the authors expect changes mdx mouse physiologic outcomes like strength, speed, weight when the drug was not having the expected biologic effect? Further, the amount of dystrophic progression normally taking place between mdx mouse ages 6-7 weeks, used for this short-term study, is likely insufficient without substantially increased power in the study . This is particularly true for pharmacologic interventions aimed at preventing “secondary disease complication” thought to account for ~30% of the total dystrophic process. Together, these problems in study design and experimental on-target effects lead to the unsurprising ineffective results which don’t accurately reflect on ONX-0914 use as a potential therapeutic for DMD, or even on its short-term use in the mdx mouse. I suggest the authors either increase the mdx mice used in the study until significan on-target ONX-0914 to Vehicle immune changes are observed. Alternatively, rewrite the manuscript such that it is clear throughout that the known ONX-0914 activities were not observed in this study. Discussion should focus on potential reasons why this activity was not observed.
Author Response
May 17, 2020
Dear Reviewers and Editor,
We appreciate the careful attention and constructive suggestions by the reviewers for our manuscript, titled “Short-term Immunoproteasome Inhibition: Performance and Muscle Phenotype in Mdx Mice.”
To be clear, we are confused, and we are at a quandary of what steps to take next and the direction to go with the manuscript. The predicament: the comments by the two reviewers are at odds with each other. For example, reviewer 1 states “Study design is appropriate and well designed to examine the study questions. The authors presented a high quality of images and results. Additionally, the conclusion from findins is well represented.” In contrast to reviewer 1, reviewer 2 first states that “The present work by Kwak et al. is well presented and uses standard MDX mouse model outcome measures for the short-term assessment of subcutaneously delivered 10 mg/kg ONX-0914.” Subsequently, the reviewer 2 follows up with an opposing statement: “the fact that the shorter-term treatment schedule and alternative delivery mechanism (SubQ), used in this study, fails to achieve similar immunoproteasome inhibition suggests a study design or experimental implementation flaw.”
Because of the opposing messages between the two reviewers and the contrasting messages within reviewer 2, we have tried to fully explain below possible reasons for the confusion by reviewer 2. We have also made changes within the manuscript.
Indeed, there is a published study by Farini et al., 2016, by which ONX-0914 was administered for a long period of time, 5 weeks, beginning when the MDX mice were 6 weeks old and ending when the mice were 11 weeks old. Foremost, we use this published study as one of our primary rationale to investigate the benefits of “short-term treatment of ONX-0914”. We cite this published paper in the manuscript, and the major finding supports the use of ONX-0914 for a long period of time as a potential treatment for secondary pathologies associated with muscular dystrophy. In contrast to the positive results with long-term treatment of ONX-0914, short-term treatment of ONX-0914 did not result in positive results; however, the treatment was not detrimental. We clearly state and conclude that ONX-0914 treatment for a short-term duration in MDX mice at 6 weeks of age does not exhibit any detrimental effect on muscle pathology nor performance; however, the treatment does not have a significant beneficial effect in the MDX muscle disease progression. We believe we interpreted our findings according to our results of a well-designed study. Unfortunately, we believe that reviewer 2 views “the lack of a positive result” as a study design flaw. We respectfully disagree, well-designed studies do not always result in positive changes.
Below, we have done our best to answer each concern of the reviewers, based on our interpretation. We used red font in the manuscript to indicate changes to the text. Below, for each one of the respective reviewer’s concern (blue font) we provide specific responses (black font.)
If there are things that we did not address/answer to your satisfaction, please let us know. We appreciate your time and effort.
Sincerely,
Drs. Dongmin Kwak, Guoxian Wei, LaDora V. Thompson, and Jong-Hee Kim
Reviewer 2
Comments and Suggestions for Authors
The present work by Kwak et al. is well presented and uses standard mdx mouse model outcome measures for the short-term assessment of subcutaneously delivered 10 mg/kg ONX-0914. The 7-week time point is within the dystrophic developmental period, where multiple rounds of degeneration/regeneration are taking place in this mouse model, prior to stabilization at 10 weeks of age. Unfortunately, there are several reasons this study will have a significantly diminished impact on the muscular dystrophy field.
First, as referenced by the authors (Farini et al. Ref#24), a longer-term study using ONX-0914 has been previously conducted in the mdx mouse model, finding several beneficial effects and clear activity on the immune presence in skeletal muscle.
Response: Yes, Farini et al. described a study using ONX-0914 for a long-term period in the MDX mouse model.
In muscle tissues following ONX-0914 treatment for 5 weeks, the mice were 11-12 weeks old at the end of the study; whereas in our study the mice were 7 weeks old. This is a five-week difference in young MDX mice between the two studies.
The Farini et al publication evaluated many tissues; the following messages were stressed about muscle: using qRT-PCR, they show a reduction in TNFα, IL1β, and IFNγ. It is difficult to determine which proteins were evaluated because the publication shows an immunoblot with only IL1β without any quantification or statistical analyses. Using muscle histochemistry, they indicate reduced signs of degeneration (hypertrophic fibers, fiber splitting, fat replacement, necrotic fibers, and fibrosis) and increased β-sarcopglycan expression. Using whole muscle contractility experimentation, they show improvement in tetanic force. To our reading, they do not provide data on LMP7 activity nor LMP7 content in muscle following the long-term ONX-0914 treatment.
In order to contrast the findings reported in the Farini et al paper, we have clearly identified the purpose of our study.
CHANGE IN MANUSCRIPT: We made the following change: “Here we aimed to explore the potential effects of short-term ONX-0914 administration (every two days for seven days) on motor performance and muscular pathology in 7-week old MDX mice, an age when the dystrophic muscles display extensive degeneration and regeneration.” This is page 2, lines 67-71.
Secondly, the fact that the shorter-term treatment schedule and alternative delivery mechanism (SubQ), used in this study, fails to achieve similar immunoproteosome inhibition suggests a study design or experimental implementation flaw. Therefore, it is difficult to draw conclusions from the lack of change between Vehicle and ONX-0914 in the study, due to a lack of on-target pharmacological activity. Due to this lack of activity, some of the conclusions discussed are unwarranted as they are based upon the presumption of ONX-0914 activity. For example, why would the authors expect changes mdx mouse physiologic outcomes like strength, speed, weight when the drug was not having the expected biologic effect?
Response: Indeed, at first we were very disappointed (like you) to see a non-significant statistical difference with the administration of ONX-0914 during the critical time period of muscle regeneration/degeneration (6-7 weeks of age) in MDX mice at both the animal motor performance and isolated muscle morphology levels. When we interpreted the results, we considered: (1) we are confident in our experimentation techniques because we were able to show significant differences between the genotypes with a sample size of 4-6 mice. (2) the muscle morphology and proteasome subunit content were consistent with the animal motor performance. Hence, we are confident that the administration of ONX-0914 for a short period of time, when there is extensive muscle degeneration/regeneration, does not result in a beneficial effect on these specific parameters.
In the data collection sequence, the motor performance characteristics and muscle tissue experimentation/analyses are collected with the mice and samples being blinded. Also, motor performance occurs before any muscle tissue analyses are initiated. See Figure1. Thus, we did not know the muscle effects prior to the motor performance effects.
In respect to the mode of ONX-0914 administration, SubQ or IP, we selected SubQ for several reasons: (1) SubQ is a physiological mode of administration in humans, (2) SubQ was used in the majority of the published papers with ONX-0914 administration, and (3) The study design reflected the SubQ 24-hour delay when assessing outcomes.
Further, the amount of dystrophic progression normally taking place between mdx mouse ages 6-7 weeks, used for this short-term study, is likely insufficient without substantially increased power in the study . This is particularly true for pharmacologic interventions aimed at preventing “secondary disease complication” thought to account for ~30% of the total dystrophic process. Together, these problems in study design and experimental on-target effects lead to the unsurprising ineffective results which don’t accurately reflect on ONX-0914 use as a potential therapeutic for DMD, or even on its short-term use in the mdx mouse. I suggest the authors either increase the mdx mice used in the study until significan on-target ONX-0914 to Vehicle immune changes are observed. Alternatively, rewrite the manuscript such that it is clear throughout that the known ONX-0914 activities were not observed in this study. Discussion should focus on potential reasons why this activity was not observed.
Response: We see the thought-process of the reviewer, and we agree with the secondary disease complications of MDX. Reflecting on the study, when we identified the sample size we used the findings from previous studies administering ONX-0914 in many inflammatory disease conditions (lines 59-62, page 2) and the Farini et al. paper. Based on the inflammatory disease studies we reviewed, ONX-0914 for a short-term treatment saw beneficial effects with sample sizes of 3-7 mice. Moreover, the Farini et al used a wide range of mice for their analyses from 3-13 mice for muscle analyses (pg1906; Figure 7.) We also selected 5-6 mice per group because we saw genotype differences; hence, we did not increase the sample size.
Based on our results in the current study, to determine a 20% difference at 80% confidence, we calculated it would be necessary to investigate 38 mice/group with these muscle characteristics (immunoproteasome subunits). Investigating both WT and MDX mice at 6 weeks of age, would require a total of 152 mice. At this time, this is not possible for us to do. Because of this required sample number and the preponderance of sample sizes used in MDX studies, we are confident we have interpreted our results correctly: ONX-0914 treatment for a short-period during the 6-7 week old MDX mouse does not cause further deterioration; however, it does not change the degeneration/regeneration process.
However, if the reviewer would like us to add the following sentence in the manuscript (line 347, page 9), we will:
POSSIBLE CHANGE IN MANUSCRIPT: Based on the current study and the specific muscle characteristics evaluated, we determined using sample size power calculations of 80% and an alpha level of 0.05, it would require 38 mice/group.
Lastly, we believe we have clearly represented the data and its interpretation in the abstract and discussion. Please see the following lines within the manuscript: 22-25, 243-267 (results section), 280-287, 326-347, and the overall conclusion at 358-359. Moreover, we provide a discussion of the down-stream signaling pathways that would be triggered by the ONX-0914 treatment, which are potential reasons (lines 309-342).
Reviewer 3 Report
The Manuscript “Short-term immunoproteasome Inhibition: Performance and Muscle Phenotype in Mdx Mice” describes the effect of short-term ONX-0914 administration (LMP7 inhibitor) on immunoproteasome content, physical performance, muscle morphology, and inflammation in both wild-type and Mdx mice. First, the study shows the genotype differences in immunoproteasome content, physical performance, CSA, CN, and inflammation between wild-type and Mdx mice. Second, the study shows the effect of short-term LMP7 inhibition on these measures in wild-type mice and mdx mice.
Although the study did not show the changes of immunoproteasome content, physical performance, muscle morphology, and inflammation after short-term ONX-0914 administration in Mdx mice, the manuscript is well written and has a logical and progressive structure. The research design and method are appropriate. The data is well presented. This study is the first step to use the short-term ONX-0914 administration as a therapeutic strategy for DMD. Overall, this manuscript merits publication.
I recommend only minor revisions to address the following points.
- Introduction, please clarify why short-term administration is necessary because long-term ONX-0914 administration has beneficial effects in mdx mice (Farini et al. 2016).
- Method, The ONX-0914 was administrated at 10mg/kg. Please clarify the volume of vehicle administration.
- Method, the Immunoblotting of α7, LMP2, and LMP7 was normalized to total protein. Usually, the immunoblotting was normalized to the content of GAPDH or α-tublin in skeletal muscle. Is there any reason to use total protein for normalization?
- Line 120, please remove the reference “(Chen et al., 2014)”.
- Line 131, please remove the reference “(Baumann et al., 2018)”.
- Result, Is there any shift on the distribution of the CSA after short-term ONX-0914 treatment in mdx mice?
- References, please check bold or not bold in author names. It is not consistent.
Author Response
"Please see the attachment."

Round 2
Reviewer 2 Report
Again, this work is well presented (well written with clear data presentation) but the study design or implementation is lacking (the known/expected pharmacological activity was not achieved). I don't believe these statements were contradictory but I'm sorry for any confusion. First the use of the terms "short-term" or "long-term" treatments are not being accurately used. In the pharmacological muscular dystrophy field using mdx mice, it is more common for a "short-term" treatment study to be conducted for 5-7 weeks between the ages of 3-10 weeks. Long-term studies more commonly are performed for up to 1 year and typically also assess mitigation of cardiac phenotypes which begin to be observable at ~9 months of age. Therefore, the Farini et al study constitutes a better powered study with similar overall design in which expected control experiments for ONX-0914 activity displayed expected results. This study also took place prior to mdx week 12 when degeneration/regeneration process stabilizes. The reviewers themselves bring up the fact that previous studies have shown positive ONX-0914 benefits within this 1-week study time frame, thus defending their study design. Unfortunately this leaves the study in a state where, as scientists, we are unsure whether ONX-0914 actually does improve this specific mouse model in this time frame (suggested conclusion of authors), if something has gone wrong with treatments (inactive compound, improper compound storage, drug delivery issues, etc.), or if study design (immune involvement in physiologic development over a 1 week period, mouse number/power/delivery mechanism, etc.) led to the lack of effect. It is possible that a significant on-target pharmacological effect would be observed with better study design (38 mice/group) but this could equally lead to positive physiologic results (strength, etc.), thus rendering the authors conclusion about efficacy incorrect. Study design encompasses all aspects of experiments, and the study design expectation should be that the compound shows on-target activity. Any result where on-target activity is not achieved must be questioned. Therefore neither of the authors conclusions are supported due to the lack of ONX-0914 activity which are inconsistent with previous reports It is highly likely that the authors assumptions are correct, however, their data does not yet support their conclusions.
Contrary to the authors beliefs, I am fine with studies lacking positive outcomes when the conclusions and data interpretation fit. As previous studies have shown pharmacological activity, in order to claim a lack of efficacy, the authors MUST either show on-target compound activity and a a lack of physiologic benefits or have a proven and justifiable reason as to why the treatments did not work in their hands. Had the authors achieved significant pharmacological activity in the absence of physiologic improvement, I believe the conclusions would be justified. However, the data in its current state, does not justify the conclusions as there are too many alternative interpretations. Perhaps the authors should consider replicating qRTPCR studies previously mentioned and used by Farini et al. (like TNFα, IL1β, and IFNγ), which may have a greater chance of being significantly altered in a 1 week period than protein levels. Alternatively, if the authors are claiming the drug does not work as previously reported, then more rigorous studies are necessary to support those conclusions. Interpreting the drug as not having any effect, when you did not find expected on-target activity in treated animals, is a MAJOR study flaw. The results can be reported as is, but the manuscript conclusions should be removed and replaced with all possible interpretations for a lack of activity as I discussed above. Basically, the study is entirely inconclusive by the scientific community in its current state.
Author Response
"Please see the attachment."
